# Unaligned Image-to-Sequence Transformation with Loop Consistency

## Abstract

We tackle the problem of modeling sequential visual phenomena. Given examples of a phenomena that can be divided into discrete time steps, we aim to take an input from any such time and realize this input at all other time steps in the sequence. Furthermore, we aim to do this *without* ground-truth aligned sequences — avoiding the difficulties needed for gathering aligned data. This generalizes the unpaired image-to-image problem from generating pairs to generating sequences. We extend cycle consistency to *loop consistency* and alleviate difficulties associated with learning in the resulting long chains of computation. We show competitive results compared to existing image-to-image techniques when modeling several different data sets including the Earth's seasons and aging of human faces.

## 1 Introduction

Image-to-image translation has gained tremendous attention in recent years. A pioneering work by (Isola et al., 2017) shows that it is possible to realize a real image from one domain as a highly realistic and semantically meaningful image in another when paired data between the domains are available. Furthermore, CycleGAN (Zhu et al., 2017) extended the image-to-image translation framework in an unpaired manner by relying on the ability to build a strong prior in each domain based off generative adversarial networks (GANs, (Goodfellow et al., 2014)) and enforcing consistency on the cyclic transformation from and to a domain. Methods (Kim et al., 2017; Liu et al., 2017) similar to CycleGAN have also been developed roughly around the same time. Since its birth, CycleGAN (Zhu et al., 2017) has become a widely adopted technique with applications even beyond computer vision (Fu et al., 2018). However, CycleGAN family models are still somewhat limited since they only handle the translation problem (directly) between two domains. Modeling more than two domains would require separate instantiations of CycleGAN between any two pairs of domains — resulting in a quadratic model complexity. A major recent work, StarGAN (Choi et al., 2018), addresses this by facilitating a fully connected domain-translation graph, allowing transformation between two arbitrary domains with a single model. This flexibility, however, appears restricted to domains corresponding to specific attribute changes such as emotions and appearance.

Within nature, a multitude of settings exist where neither a set of pairs nor a fully-connected graph are the most natural representations of how one might proceed from one domain to another. In particular, many natural processes are sequentialand therefore the translation process should reflect this. A common phenomena modeled as an image-to-image task is the visual change of natural scenes between two seasons (Zhu et al., 2017), e.g., Winter and Summer. This neglects the fact that nature first proceeds to Spring after Winter and Fall after Summer and therefore the pairing induces a very discontinuous reflection of the underlying process. Instead, we hope that by modeling a higher resolution discretization of this process, the model can more realistically approach the true model while reducing the necessary complexity of the model.

It is difficult to obtain paired data for many image-to-image problems. Aligned sequential are even more difficult to come by. Thus, it is more plausible to gather a large number of examples from each step (domain) in a sequence without correspondences between the content of the examples. Therefore, we consider a setting similar to unpaired image-to-image transformation where we only have access to unaligned examples from each time step of the sequence being modeled. Given an example from an arbitrary point in the sequence, we then generate an *aligned* sequence over all other time steps — expecting a faithful realization of the image at each step. The key condition that

required is that after generating an entire loop (returning from the last domain to the input domain), one should expect to return to the original input. This is quite a weak condition and promotes model flexibility. We denote this extension to the cycle consistency of (Zhu et al., 2017) as *loop consistency* and therefore name our approach as *Loop-Consistent Generative Adversarial Networks (LoopGAN)*. This is a departure from many image-to-image approaches that have very short (usually length 2) paths of computation defining what it means to have gone "there and back", e.g. the ability to enforce reconstruction or consistency. Since we do not have aligned sequences, the lengths of these paths for LoopGAN are as large as the number of domains being modeled and require different approaches to make learning feasible. These are not entirely different from the problems that often arise in recurrent neural networks and we can draw similarities to our model as a memory-less recurrent structure with applied to images.

We apply our method to the sequential phenomena of human aging (Zhang & Qi, 2017) and the seasons of the Alps (Anoosheh et al., 2018) with extensive comparisons with baseline methods for image-to-image translation. We also present additional results on gradually changing azimuth angle of chairs and gradual change of face attributes to showcased the flexibility of our model. We show favorable results against baseline methods for image-to-image translation in spite of allowing for them to have substantially larger model complexity.

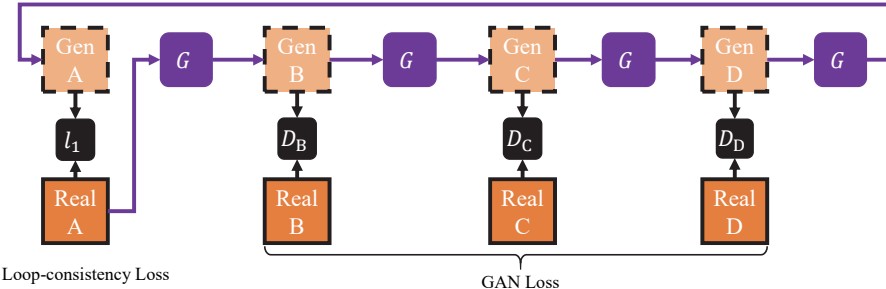

Figure 1: LoopGAN framework: for simplicity, only a single loop starting at one real domain in a four-domain problem is illustrated here. All four steps share a single generator $G$, parameterized by the step variable. When training $G$, our objective function actually consists of four loops including $A \to B \to C \to D \to A$, $B \to C \to D \to A \to B$, $C \to D \to A \to B \to C$, and $D \to A \to B \to C \to D$. This is consistent with how CycleGAN is trained where two cycles are included.

## 2 RELATED WORK

**Generative Adversarial Networks**    Generative adversarial networks (GANs, (Goodfellow et al., 2014)) implicitly model a distribution through two components, a generator $G$ that transforms a sample from a simple prior noise distribution into a sample from the learned distribution over observable data. An additional component known as the discrimintor $D$, usually a classifier, attempts to distinguish the generations of $G$ with samples from the data distribution. This forms a minimax game from which both $G$ and $D$ adapt to one another until some equilibrium is reached.

**Unpaired Image-to-Image Transformation**    As an extension to the image-to-image translation framework (pix2pix, (Isola et al., 2017)), (Zhu et al., 2017) proposed CycleGAN which has a similar architecture as in (Isola et al., 2017) but is able to learn transformation between two domains without paired training data. To achieve this, CycleGAN simultaneously train two generators, one for each direction between the two domains. Besides the GAN loss enforced upon by domain-wise discriminators, the authors proposed to add a cycle-consistency loss which forces the two generators to be reversible. Similar to pix2pix, this model aims at learning a transformation between *two* domains and cannot be directly applied in multi-domain setting that involves more than two domains. Concurrent to CycleGAN, (Liu et al., 2017) proposed a method named UNIT that implicitly achieves alignment between two domains using a VAE-like structure where both domains share a common latent space. Furthermore, StarGAN ((Choi et al., 2018)) proposed an image-to-image translation model for multiple domains. A single network takes inputs defining the source image and desired domain transformation, however, it has been mainly shown to be successful for the domains consisting of facial attributes and expressions.

**Multi-Modal Transformation**   The problem of learning non-deterministic multi-modal transformation between two image domains has made progress in recent years ((Huang et al., 2018; Liu et al., 2018)). The common approach that achieves good performance is to embed the images for both domains into a shared latent space. At test time, an input image in the source domain is first embedded into the shared latent space and decoded into the target domain conditioned on a random noise vector. These models avoid one-to-one deterministic mapping problem and are able to learn different transformations given the same input image. However, these models are developed exclusively for two-domain transformation and cannot be directly applied to problems with more than two domains.

**Style Transfer**   A specific task in image-to-image transformation called style transfer is broadly defined as the task of transforming a photo into an artistic style while preserving its content (Gatys et al., 2015; Johnson et al., 2016). Common approaches use a pre-trained CNN as feature extractor and optimize the output image to match low-level features with that of style image and match high-level features with that of content image (Gatys et al., 2015; Johnson et al., 2016). A network architecture innovation made popular by this field known as AdaIn (Huang & Belongie, 2017; Dumoulin et al., 2017) combines instance normalization with learned affine parameters. It needs just a small set of parameters compared to the main network weights achieve different style transfers within the same network. It also shows great potential in improving image quality for image generation (Karras et al., 2019) and image-to-image transformation (Huang et al., 2018).

**Face Aging**   Generating a series of faces in different ages given a single face image has been widely studied in computer vision. State-of-the-art methods (Zhang & Qi, 2017; Palsson et al., 2018) use a combination of pre-trained age estimator and GAN to learn to transform the given image to different ages that are both age-accurate and preserve original facial structure. They rely heavily on a domain-specific age estimator and thus have limited application to the more general sequential image generation tasks that we try to tackle here.

**Video Prediction**   Video prediction attempts to predict some number of future frames of a video based on a set of input frames (Shi et al., 2015; Vondrick et al., 2016). Full videos with annotated input frames and target frames are often required for training these models. A combination of RNN and CNN models has seen success in this task (Srivastava et al., 2015; Shi et al., 2015). Predictive vision techniques (Vondrick et al., 2016; Vondrick & Torralba, 2017; Wang et al., 2019) that use CNN or RNN to generate future videos also require aligned video clips in training. A recent work (Gupta et al., 2018) added a GAN as an extra layer of supervision for learning human trajectories. At a high level, video prediction can be seen as a supervised setting of our unsupervised task. Moreover, video prediction mostly aims at predicting movement of objections rather than transformation of a still object or scene which is the focus of our task.

## 3   METHOD

We formulate our method and objectives. Consider a setting of $n$ domains, $X_1, \ldots, X_n$ where $i < j$ implies that $X_i$ occurs temporally before $X_j$. This defines a sequence of domains. To make this independent of the starting domain, we additionally expect that can translate from $X_n$ to $X_1$ — something a priori when the sequence represents a periodic phenomena. We define a *single* generator $G(x, i)$ where $i \in \{1, \ldots, n\}$ and $x \in X_i$. Then, a translation between two domains $X_i$ and $X_j$ of an input $x_i \in X_i$ is given by repeated applications of $G$ in the form of $G^{\|j-i\|}(x_i, i)$ (allowing for incrementing the second argument modulo $n + 1$ after each application of $G$). By applying $G$ to an input $n$ times, we have formed a direct loop of translations where the source and target domains are equal. While we use a single generator, we make use of $n$ discriminators $\{D_i\}_{i=1}^n$ where $D_i$ is tasked with discriminating between a translation from any source domain to $X_i$. Since we are given only samples from each domain $X_i$, we refer to each domain $X_i = \{x_j\}_{j=1}^{N_i}$ as consisting of $N_i$ examples from the domain $X_i$ with data distribution $p_{data}(x_i)$.

## 3.1 ADVERSARIAL LOSS

Suppose $x_i \sim p_{data}(x_i)$. Then we expect that for all other domains $j$, $G^{||j-i||}(x_i, i)$ should be indistinguishable under $D_j$ from (true) examples drawn from $p_{data}(x_j)$. Additionally, each $D_j$ should aim to minimize the ability for $G$ to generate examples that it cannot identify as fake. This forms the adversarial objective for a specific domain as:

$$\mathcal{L}_{GAN}(G, D_i) = \mathop{\mathbb{E}}_{x_i \sim p_{data}(x_i)} [\log D_i(x_i)] + \sum_{j \neq i} \mathop{\mathbb{E}}_{x_j \sim p_{data}(x_j)} [\log(1 - D_i(G^*(x_j)))]$$

where $G^*$ denotes iteratively applying $G$ until $x_j$ is transformed into domain $X_i$, i.e. $||j - i||$ times. Taking this over all possible source domains, we get an overall adversarial objective as:

$$\mathcal{L}_{GAN}(G, D_1, \dots, D_n) = \mathop{\mathbb{E}}_{i \sim q(i)} \left[ \mathop{\mathbb{E}}_{x_i \sim p_{data}(x_i)} [\log D_i(x_i)] + \sum_{j \neq i} \mathop{\mathbb{E}}_{x_j \sim p_{data}(x_j)} [\log(1 - D_i(G^*(x_j)))] \right]$$

where $q(i)$ is a prior on the set of domains, eg. uniform.

## 3.2 LOOP CONSISTENCY LOSS

Within (Zhu et al., 2017), an adversarial loss was supplemented with a cycle consistency loss that ensured applying the generator from domain $A$ to domain $B$ followed by applying a *separate* generator from $B$ to $A$ acts like an identity function. However, LoopGAN only has a single generator and supports an arbitrary number of domains. Instead, we build a loop of computations by applying the generator $G$ to a source image $n$ times (equal to the number of domains being modeled). This constitutes loop consistency and allows us to reduce the set of possible transformations learned to those that adhere to the consistency condition. Loop consistency takes the form of an $L_1$ reconstruction objective for a domain $X_i$ as:

$$\mathcal{L}_{Loop}(G, X_i) = \mathop{\mathbb{E}}_{x_i \sim p(x_i)} ||x_i - G^n(x_i, i)||_1$$

## 3.3 FULL OBJECTIVE

The combined loss of LoopGAN over both adversarial and loop-consistency losses can be written as:

$$\mathcal{L}(G, D_1, \dots, D_n, X_1, \dots, X_n) = \mathcal{L}_{GAN}(G, D_1, \dots, D_n) + \lambda \mathbb{E}_{i \sim q(i)} [\mathcal{L}_{Loop}(G, X_i)]]$$

$$= \mathbb{E}_{i \sim q(i)} \left[ \mathbb{E}_{x_i \sim p_{data}(x_i)} \left[ \log D_i(x_i) \right] + \sum_{j \neq i} \mathbb{E}_{x_j \sim p_{data}(x_j)} [\log (1 - D_i(G^*(x_j)))] \right.$$

$$\left. + \lambda \mathbb{E}_{x_i \sim p_{data}(x_i)} ||x_i - G^n(x_i)||_1 \right] \tag{1}$$

where $\lambda$ weighs the trade-off between adversarial and loop consistency losses.

An example instantiation of our framework for one loop in a four-domain problem is shown in Figure 1.

# 4 IMPLEMENTATION

## 4.1 NETWORK ARCHITECTURE

We adopt the network architecture for style transfer proposed in (Johnson et al., 2016) as our generator. This architecture has three main components: a down-sampling module $Enc(x)$, a sequence of residual blocks $T(h, i)$, and an up-sampling module $Dec(h)$. The generator $G$ therefore is the composition $G(x, i) = Dec(T(Enc(x), i))$ where the dependence of $T$ on $i$ only relates to the step-specific AdaIN parameters (Huang & Belongie, 2017) while all other parameters are independent of

$i$. Following the notations from (Johnson et al., 2016; Zhu et al., 2017), let c7-k denote a $7 \times 7$ Conv-ReLU layer with k filters and stride 1, dk denote a $3 \times 3$ Conv-ReLU layer with k filters and stride 2, Rk denote a residual block with two $3 \times 3$ Conv-AdaIn-ReLU layers with k filters each, uk denotes a $3 \times 3$ fractional-strided-Conv-LayerNorm-ReLU layer with k filters and stride $\frac{1}{2}$. The layer compositions of modules are down-sampling: c7-32, d64, d128; residual blocks: R128 $\times$ 6; up-sampling: u128, u64, c7-3. We use the PatchGAN discriminator architecture as (Zhu et al., 2017): c4-64, c4-128, c4-256, c4-1, where c4-k denotes a $4 \times 4$ Conv-InstanceNorm-LeakyRelu(0.2) layer with k filters and stride 2.

### 4.2 RECURRENT TRANSFORMATION

Suppose we wish to translate some $x_i \in X_i$ to another domain $X_j$. A naive approach would formulate this as repeated application of $G$, $|j - i|$ times. However, referencing our definition of $G$, we can unroll this to find that we must apply $Enc$ and $Dec$ $j - i$ times throughout the computation. However, $Enc$ and $Dec$ are only responsible for bringing an observation into and out of the space of $T$. This is not only a waste of computation when we only require an output at $X_j$, but it has serious implications for the ability of gradients to propagate through the computation. Therefore, we implement $G(x_i, i)$ as: a single application of $Enc(x_i)$, $j - i$ applications of $T(h)$, and a single application of $Dec(h)$. $T$ is applied *recurrently* and the entire generator is of the form:

$$G(x_i, i) = Dec(T^{|j-i|}(Enc(x_i)))$$

We show in our ablation studies that this re-formulation is critical to the learning process and the resulting quality of the transformations learned. Additionally, $T(h, i)$ is given a a set of separate, learnable normalization (AdaIN (Huang & Belongie, 2017)) parameters that it selects based off of $i$ with all other parameters of $T$ being stationary across time steps. The overall architecture is shown in Figure 2.

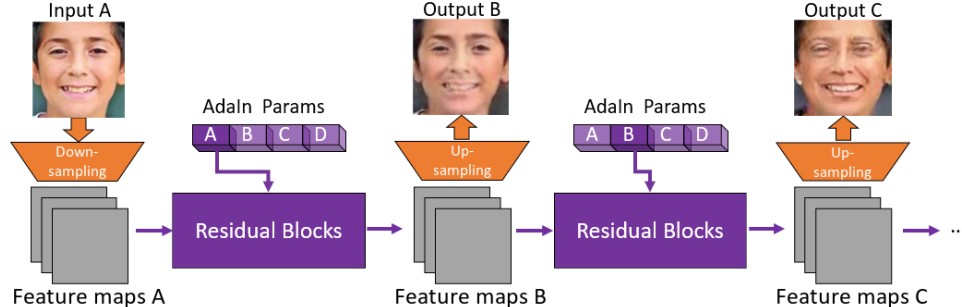

Figure 2: LoopGAN network. All modules share parameters.

### 4.3 TRAINING

For all datasets, the loop-consistency loss coefficient $\lambda$ is set to 10. We use Adam optimizer ((Kingma & Ba, 2014)) with the initial learning rate of 0.0002, $\beta_1 = 0.5$, and $\beta_2 = 0.999$. We train the face aging dataset and Alps seasons dataset for 50 epochs and 70 epochs respectively with initial learning rate and linearly decay learning rate to 0 for 10 epochs for both datasets.

## 5 EXPERIMENTS

We apply LoopGAN to two very different sequential image generation tasks: face aging and chaging seasons of scenery pictures. Baselines are built with two bi-domain models, CycleGAN (Zhu et al., 2017) and UNIT (Liu et al., 2017) and also a general-purpose multi-domain model StarGAN (Choi et al., 2018). We are interested in the sequential transformation capabilities of separately trained bi-domains compared to LoopGAN. Therefore, for each of the two bi-domains models, we train a separate model between every pair of sequential domains, i.e. $X_i$ and $X_{i+1}$ and additionally

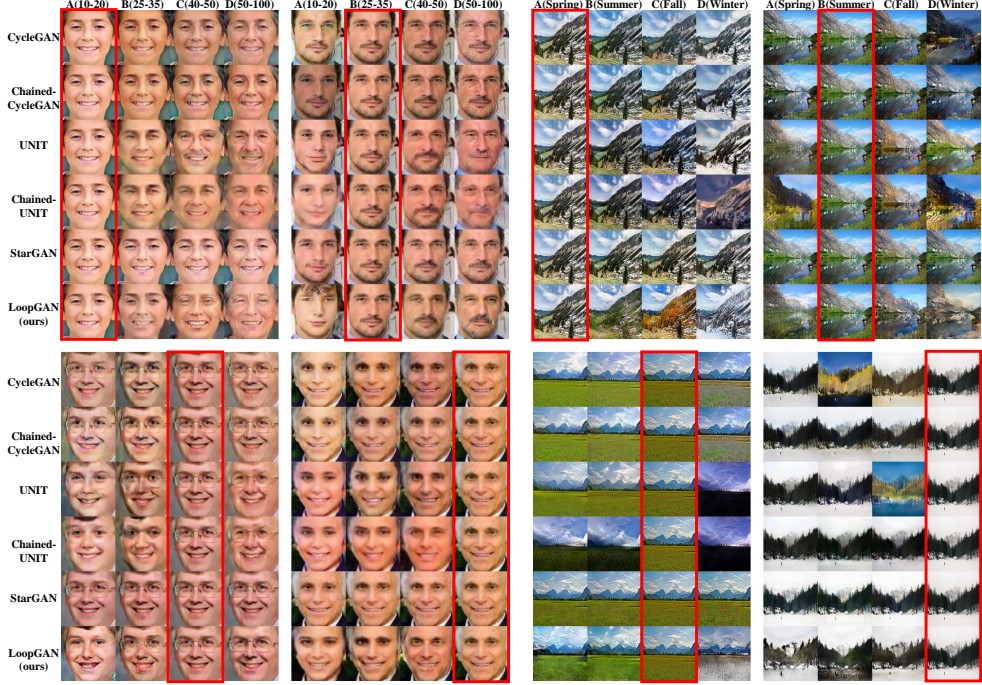

Figure 3: Face Aging and Alps seasons change with LoopGAN compared to baselines. Input real images are highlighted with rectangles (viewed in color).

train a model between every pair (not necessarily sequential) domains $X_i$ and $X_j$ ($i \neq j$). The first approach allows us to build a baseline for sequential generation by *chaining* the (separately learned) models in the necessary order. For instance, if we have four domains: A, B, C, D, then we can train four separate CycleGAN (or UNIT) models: $G_{AB}, G_{BC}, G_{CD}, G_{DA}$ and correctly compose them to replicate the desired sequential transformation. Additionally, we can train direct versions e.g. $G_{AC}$ of CycleGAN (or UNIT) for a more complete comparison against LoopGAN. We refer to composed versions of separately trained models as *Chained-CycleGAN* and *Chained-UNIT* depending on the base translation method used. Since StarGAN ((Choi et al., 2018)) inherently allows transformation between any two domains, we can apply this in a chained or direct manner without any additional models needing to be trained.

## 5.1 FACE AGING

We adopt the UTKFace dataset (Zhang & Qi, 2017) for modeling the face aging task. It consists of over 20,000 face-only images of different ages. We divide the dataset into four groups in order of increasing age according to the ground truth age given in the dataset as A consisting of ages from 10-20, B containing ages 25-35, C containing ages 40-50, and D containing ages 50-100. The number of images for each group are 1531, 5000, 2245, 4957, respectively, where a 95/5 train/test split is made. The results of LoopGAN generation are shown in on the left side in Figure 3.

LoopGAN shows advantage over baseline models in two aspects. The overall facial structure is preserved which we believe is due to the enforced loop consistency loss. Moreover, LoopGAN is able to make more apparent age changes compared to the rest of baseline models.

In order to quantitatively compare the amount of age change between models, we obtain an age distribution of generated images by running a pre-trained age estimator DEX (Rothe et al., 2015). The estimated age distributions of generated images (from input test images) are compared against those of the train images in Figure 4. The age distribution of LoopGAN generated images is closer to that of the train images across all four age groups when compared to the baseline models — suggesting that it more faithfully learns the sequential age distribution changes of the training data.

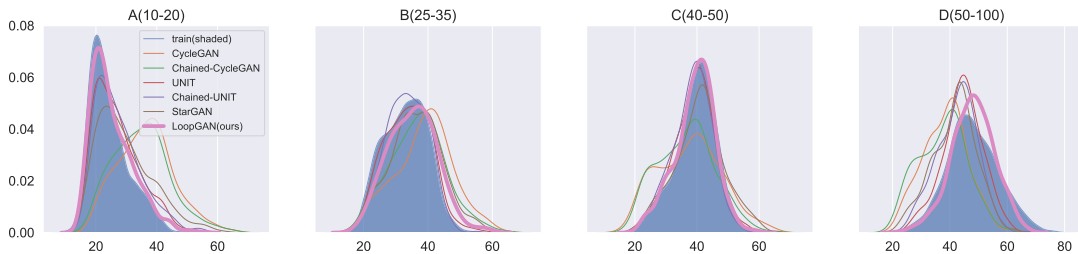

Figure 4: Comparing estimated age distribution between model generated images and train images.

## 5.2 Changing Seasons

We use the collected scenery photos of Alps mountains of four seasons from (Anoosheh et al., 2018). They are ordered into a sequence starting from Spring (A), to Summer (B), Fall (C), and Winter (D). They each have approximately 1700 images and are divided into 95/5% training and test set.

We show the results in Figure 3. Overall, LoopGAN is able to make drastic season change while maintaining the overall structure of the input scenery images. To further quantify the generation results, we conducted a user study with Amazon Mechanical Turk (AMT) Table 1 which shows that LoopGAN generations are preferred by human users.

Table 1: AMT user study results. 20 users are given each set in Figure 3 above and asked to choose the best generation sequence. LoopGAN generations are preferred by human users over baseline models.

| Model | Face Aging | Season Change | Overall |
|---|---|---|---|
| CycleGAN | 7.50% | 11.25% | 9.375% |
| Chained-CycleGAN | 17.50% | 13.75% | 15.625% |
| UNIT | 12.50% | 16.25% | 14.375% |
| Chained-UNIT | **27.50%** | 20.00% | 23.750% |
| StarGAN | 11.25% | 10.00% | 10.625% |
| LoopGAN (ours) | 23.75% | **28.75%** | **26.250%** |

## 5.3 Additional Datasets

To showcase the universality of our model, we apply LoopGAN to two additional datasets in four different sequential transformation tasks: chairs with different azimuth angles, and gradual change of face attributes in degree of smiling, gender feature, and hair color. The chairs dataset (Aubry et al., 2014) comes with ground truth azimuth angle and is divided into four sets each containing chairs facing a distinct direction. To obtain linear manifolds for the face attributes, we train a binary classifier with 0/1 labels available for each attribute (Liu et al.) and use the predicted probability to determine the position of an image on an attribute manifold. The results are shown in Figure 5.

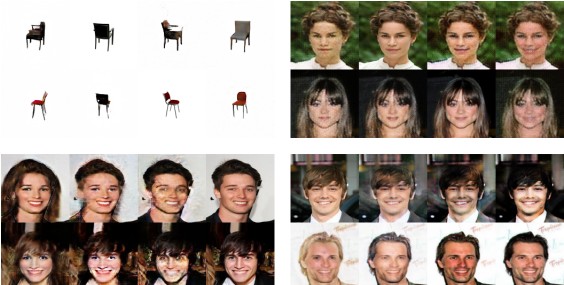

Figure 5: Additional datasets. Top row: (left) changing azimuth of a chair, (right) increasing degree of smile. Bottom row: (left) gradually changing gender features, (right) gradually changing hair color.

## 6 Ablation studies

We experiment with several network architecture variations and investigate their effect on generation quality. First, attention mechanisms have proven to be useful in GAN image generation (Zhang et al., 2018). We added attention mechanim in both space and time dimension (Wang et al., 2018), however we found that the network struggles to generate high quality image after adding this type of attention mechanism. We also noticed that (Huang et al., 2018) mentioned that for down-sampling, it is better to use no normalization to preserve information from input image, and for up-sampling it is better to use layer-normalization for faster training and higher quality. We applied these changes and found that they indeed help the network produce better results. The results under these variations are shown in Figure 6.a (first three rows).

| Model | Parameter Count |
|---|---|
| CycleGAN | 94.056 M * |
| Chained-CycleGAN | 62.704 M * |
| UNIT | 133.680 M * |
| Chained-UNIT | 89.120 M * |
| StarGAN | 8.427 M |
| LoopGAN(ours) | 11.008 M |

(a)                                                    (b)

Figure 6: (a). Ablation study for the architecture changes. (b) Model size comparison. * Note that the parameter count for vanilla and chained versions of bi-domain models (CycleGAN, Chained-CycleGAN, UNIT, and Chained-UNIT) are totals of separate pair-wise generators that together facilitate sequence generation.

Moreover, we show the importance of the recurrent form of $T(h)$ discussed in Section 4.2. We compare the choice to invoke $Enc$ and $Dec$ at each time step versus applying them once with some number of recurrent applications of $T$ in Figure 6.a (last row) and show the poor quality observed when performing the loop naively.

Lastly, we calculate the parameter count of generator networks compared in the face aging and season change experiments above and show that our final generator network architecture is parameter-efficient compared to baseline models in Figure 6.b.

For completeness, we also include a selection of failure cases in table in Figure 7.

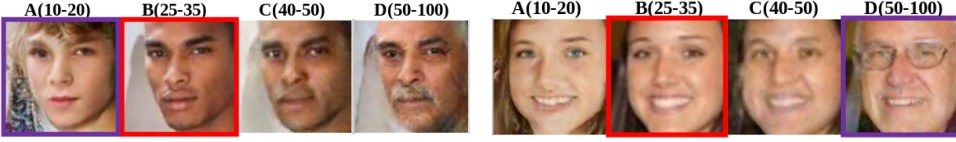

(a) Failure case 1.                                   (b) Failure case 2.

Figure 7: Two failure cases. Input images and failure generations are highlighted respectively in red and purple (viewed in color). In both cases, the highlighted generated images (the first column in (a) and the last column in (b)) bear some semantic dissimilarity to the input images. It seems that sometimes the network overfit to more drastic transformations that only preserve overall facial structure and orientation but neglects all other features.

## 7 Conclusion

We proposed an extension to the family of image-to-image translation methods when the set of domains corresponds to a sequence of domains. We require that the translation task can be modeled as a consistent loop. This allows us to use a shared generator across all time steps leading to significant efficiency gains over a nave chaining of bi-domain image translation architectures. Despite this, our architecture shows favorable results when compared with the classic CycleGAN family algorithms.

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
