# OpenReview forum: "Unaligned Image-to-Sequence Transformation with Loop Consistency"
_ICLR.cc/2020/Conference — Reject_

### Official Review · AnonReviewer3 · 2019-10-23
**Official Blind Review #3**

**Rating:** 1

**Review:**

This paper introduces the LoopGan, which aims to enforce consistency in a sequential set of images but without aligned pairs.  The prototypical example used in the paper is a network to transform seasonal images, where Summer -> Fall -> Winter -> Spring -> Summer -> etc.

I suggest rejecting the paper.

While the idea has some merit, it is an interesting premise to train a recurrent generator to create images in multiple domains, I feel as though the experiments are quite lacking.  The primary evidence the method works is Figure 3, for which we see only 4 examples of the method on two tasks.  Looking at the results they don't look all that great and the proposed method is hard to identify as the best.

The idea doesn't strike me as particular innovative, feeling like a natural extension of the prior work listed.

Table 1 shows the results of a user study with 20 users.  With only 20 users rating 4 examples of each method we get very little power to resolve the best method, and the papers proposed method is selected in aggregate only 26% of the time out of 6 methods, this is hardly a clear winner.

Figure 4 showing the imputed ages for the generated images doesn't seem that strong of evidence the proposed method is that much better either.  The histograms are difficult to compare by eye, perhaps the paper should report computed estimates of the kl divergence between the two?

The ablation studies of section 6 are out of place.  The paper does not compare against the incremental removal of its' proposed loss, it simply reports results of alternative architectures.

**Experience Assessment:**

I do not know much about this area.

**Review Assessment: Checking Correctness Of Derivations And Theory:**

N/A

**Review Assessment: Checking Correctness Of Experiments:**

I assessed the sensibility of the experiments.

**Review Assessment: Thoroughness In Paper Reading:**

I read the paper at least twice and used my best judgement in assessing the paper.

---

### Official Review · AnonReviewer2 · 2019-10-23
**Official Blind Review #2**

**Rating:** 3

**Review:**

This paper proposes to extend CycleGAN by replacing a bidirectional cycle with a "loop" or a sequence of transformations across gradually changing image domains (sequential transformation). This is achieved with a simple modification to CycleGAN training where the training involves additional steps to create a loop. A number of experimental results are shown on specific sequential datasets, comparing to a number of other baselines. On the chosen datasets and tasks, the proposed architecture provides good results over the baselines. However, I would have liked to see ablation studies showing that they achieve the same results as CycleGAN when training on 2 domains. The paper's technical novelty is limited to this simple loop extension and generator sharing. As such, while the experimental results are reasonable, I find this lack of any technical novelty a negative factor.

**Experience Assessment:**

I have published in this field for several years.

**Review Assessment: Checking Correctness Of Derivations And Theory:**

N/A

**Review Assessment: Checking Correctness Of Experiments:**

I carefully checked the experiments.

**Review Assessment: Thoroughness In Paper Reading:**

N/A

---

### Official Review · AnonReviewer1 · 2019-10-29
**Official Blind Review #1**

**Rating:** 3

**Review:**

This paper proposes loop consistency on the base of cycle consistency and alleviate difficulties associated with learning in the resulting long chains of computation. Like CycleGAN, the method proposed in this paper does not require data to be specifically matched.

This paper still has the following problems：
1）For the time series images, the reason for processing with a single generator is not clearly stated. In CycleGAN, the two transformations use two different generators, respectively, and use the cycle consistency loss to force the results to converge. In the examples presented in this paper, such as Face Aging, it isn't a process that can be reversed. This paper does not explain why the use of a single generator has been effective.
2）To be more convincing, this article needs to be tested on more baselines. The indicators provided in this article are not objective enough.
3）This article also does not give the training computational complexity and testing time cost of the proposed method.


**Experience Assessment:**

I have published one or two papers in this area.

**Review Assessment: Checking Correctness Of Derivations And Theory:**

I did not assess the derivations or theory.

**Review Assessment: Checking Correctness Of Experiments:**

I did not assess the experiments.

**Review Assessment: Thoroughness In Paper Reading:**

I read the paper at least twice and used my best judgement in assessing the paper.

---

### Decision · Program_Chairs · 2019-12-19

**Decision:**

Reject

**Comment:**

The main concern raised by the reviewers is limited experimental work, and there is no rebuttal.